# MTMR7 regulates human spermatogonial stem cells proliferation and migration via targeting FLNB

Nianchao Zhou[1◉], Tiantian Wu[2◉], Yi Yu[3◉], Wenxin Gao[4], Bing Jiang[1], Haoyue Hu[1], Xiaoyan Huang[4], Cong Shen [2]*, Yibo Wu[1]*, Tingting Gao[5]*

**1** Human Reproductive and Genetic Center, Affiliated Hospital of Jiangnan University, Wuxi, China, **2** State Key Laboratory of Reproductive Medicine and Offspring Health, Center for Reproduction and Genetics, The Affiliated Suzhou Hospital of Nanjing Medical University, Suzhou Municipal Hospital, Gusu School of Nanjing Medical University, Suzhou, China, **3** Reproductive Medicine Center, The First Affiliated Hospital of Ningbo University, Ningbo, China, **4** State Key Laboratory of Reproductive Medicine and Offspring Health, Department of Histology and Embryology, School of Basic Medical Sciences, Nanjing Medical University, Nanjing, China, **5** Changzhou Medical Center, Changzhou Maternal and Child Health Care Hospital, Nanjing Medical University, Changzhou, China

◉ These authors contributed equally to this work.
* congshen@njmu.edu.cn (CS); moliaty@aliyun.com (YW); 960182162@qq.com (TG)

## Abstract

In the testes, spermatogonial stem cells (SSCs) maintain normal spermatogenesis through the dual potential of self-renewal and differentiation, which is essential for male fertility. Myotubularin-associated protein 7 (MTMR7), as a vital member of MTMR family with phosphatase activity, is involved in a variety of membrane transport processes through regulating the levels of phosphoinositides (PIPs). Our earlier research demonstrated that MTMR7 controls the cell cycle and maintains homeostasis in mouse SSCs by inhibiting PI3K/AKT signaling. However, its role in human SSCs has not been reported. This study found that knocking down MTMR7 increased the proliferation and migration of human SSCs, whereas MTMR7 overexpression inhibited these processes. Through mass spectrometry and immunoprecipitation, we identified filamin B (FLNB) as an interacting protein of MTMR7, and MTMR7 is required for FLNB ubiquitination and subsequent degradation. Further validation using immunofluorescence confirmed the involvement of the downstream β-catenin signaling. Altogether, this study is the first to demonstrate that MTMR7 regulates β-catenin expression and inhibits human SSCs proliferation and migration by mediating the ubiquitination and degradation of FLNB. These findings may offer new therapeutic strategies and target gene loci for treating male infertility caused by SSCs dysfunction.

**Data availability statement:** All relevant data are within the paper and its Supporting information files.

**Funding:** This work was supported by the National Natural Science Foundation of China (82201762), the Suzhou Gu Su Health Talent Research Project (GSWS2023056), and the Top Talent Support Program for Young and Middle-Aged People of Wuxi Health Committee (BJ2020047). The funders had no role in study design, data collection and analysis, decision to publish, or preparation of the manuscript.

**Competing interests:** The authors have declared that no competing interests exist.

## Introduction

Male infertility is a complex condition affecting approximately 7% of men worldwide and arises from a combination of genetic, environmental, lifestyle, and pathological factors [1]. Beyond its impact on reproduction, male infertility may also reflect broader health issues, as evidenced by studies showing a threefold increased cancer risk among azoospermic men [2]. Azoospermia, defined by the complete absence of spermatozoa in the ejaculate, is one of the most severe forms of male infertility and accounts for approximately 15% of cases [3,4]. Based on pathological features and underlying etiology, azoospermia is classified into two major types: obstructive azoospermia (OA), resulting from blockage of the vas deferens, and non-obstructive azoospermia (NOA), which arises from intrinsic defects in spermatogenesis. NOA accounts for approximately 70% of azoospermia cases and is primarily attributed to impaired function of spermatogonial stem cells (SSCs) or dysregulation of their niche microenvironment [5,6].

SSCs are adult tissue stem cells located in the testes that are fundamental to spermatogenesis, and their dysfunction can lead to male infertility [7,8]. SSCs possess dual potential: they can either self-renew to produce new SSCs or differentiate into mature spermatozoa [9]. Maintaining a balance between these processes is crucial for normal spermatogenesis. Earlier research has shown that glial cell line-derived neurotrophic factor (GDNF) is pivotal in controlling both the proliferation and differentiation of mouse SSCs. High concentrations of GDNF promote SSCs proliferation, while low concentrations induce SSCs differentiation [10–12]. Moreover, fibroblast growth factor 2 (FGF2) and retinoic acid (RA) have been recognized as crucial in guiding the developmental decisions of mouse SSCs [13–15]. However, the molecular mechanisms controlling the development of human SSCs remain largely unknown and differ significantly from those in mice [16]. For instance, in addition to the commonly expressed markers PLZF and GPR125, human SSCs uniquely express MAGE-A4 and CD133, while mouse SSCs are marked by CD29, KIT, and POU5F1 [17,18]. The CXCR4–CXCL12 signaling pathway, essential for SSCs homing to the niche, also differs between species: in humans, CXCL12 is mainly expressed by Leydig cells, whereas in mice, it is predominantly produced by Sertoli cells [19]. These differences underscore the need for species-specific studies to elucidate the regulatory mechanisms governing human SSCs function, which is essential for developing targeted therapies for male infertility.

The myotubularin (MTM) and myotubularin-related protein (MTMR) families comprise a total of 15 members, most of which possess phosphatase activity and dephosphorylate phosphatidylinositol 3-phosphate (PtdIns3P) and PtdIns(3,5)$P_2$ to generate PtdIns and PtdIns5P, respectively [20]. These phosphoinositides (PIPs) act as second messengers in cells and are involved in various cellular processes, including cell signaling, protein translocation, cytoskeletal organization, and membrane transport [21]. As regulators of PIPs levels, MTMRs are broadly expressed in different tissues and organs and play crucial roles in cell membrane transport and endocytosis [22]. Notably, two members, MTMR5 and MTMR2, are prominently expressed in both supporting cells and germ cells within the testis, and their knockdown results in spermatogenic defects [23,24].

As a key member of the MTMR family, myotubularin-related protein 7 (MTMR7) differs from other members that primarily dephosphorylate PtdIns3P and PtdIns(3,5)P$_2$, as it exhibits a stronger catalytic preference for its specific substrate, inositol 1,3-bisphosphate (Ins(1,3)P$_2$) [25]. This unique enzymatic activity may underlie its distinct biological functions. MTMR7 has been shown to significantly inhibit proliferation across a range of cell types, such as gastric cancer cells, vascular smooth muscle cells, colorectal cancer cells and myoblasts [26–29]. In the context of male reproduction, our previous studies demonstrated that MTMR7 is specifically expressed in SSCs in neonatal mouse testes and contributes to the maintenance of cell cycle homeostasis by negatively regulating the PI3K/AKT signaling pathway [30]. These findings suggest that MTMR7 may also serve as a potential biomarker in human SSCs. However, given the phenotypic and regulatory differences between human and mouse SSCs, the precise role and underlying molecular mechanisms of MTMR7 in human SSCs remain to be elucidated.

In this research, we examined how MTMR7 influences human SSCs and explored the molecular mechanisms involved. Our findings reveal that MTMR7 could interact with filamin B (FLNB) and thereby promoting the ubiquitination and degradation of FLNB. This interaction leads to a reduction of the downstream protein, β-catenin, ultimately impacting the function of SSCs. These results not only elucidate the molecular mechanisms regulating the fate of human SSCs but also reveal potential new targets for therapeutic interventions in male infertility associated with SSCs dysfunction.

## Materials and methods

### Sample collection

Human testicular tissues were obtained from a 34-year-old healthy male donor with normal spermatogenesis, whose partner was receiving conventional in vitro fertilization (IVF) treatment at Suzhou Municipal Hospital. The tissue samples were collected during the recruitment period from 30 October 2023–1 November 2023. Paraffin sections were prepared by rinsing tissues with phosphate-buffered saline (PBS) and fixing in 4% paraformaldehyde [31,32]. This study was approved by the Ethics Committee of Suzhou Municipal Hospital (No.2023005) and a written informed consent from the donor was obtained in accordance with the Declaration of Helsinki.

### Cell culture

The human SSC line was generously provided by Professor Zuping He (Hunan Normal University, China). These cells express characteristic SSCs markers, including GPR125, GFRA1, and PLZF, as confirmed by immunocytochemistry [33,34]. The cells were cultured in DMEM/F-12 medium (Gibco) to support their growth, with additional supplements including 10% fetal bovine serum (FBS, ScienCell, USA) to provide essential nutrients, and 1% penicillin-streptomycin solution to prevent bacterial contamination.

### si-RNA, plasmids and transfection reagents

When the cell density reached approximately 70%, overexpression plasmids pcDNA3.1-Flag-MTMR7 and an empty vector (Sangon, Shanghai, China) were introduced into SSCs using the X-treme GENE HP DNA transfection reagent (Mannheim, Germany). For plasmid transfections, 10 μg DNA was used per 10-cm dish and 2 μg per well of a 6-well plate. Separately, siRNAs targeting MTMR7 and FLNB (GenePharma, Suzhou, China) were transfected into SSCs using Lipofectamine 2000 (Invitrogen, USA), with 10 μl siRNAs per well in a 6-well plate [35–37]. To minimize off-target effects, two independent siRNAs were designed for each gene, and only consistent results from both were considered valid. The sequences targeted by the siRNAs were as follows:

si-MTMR7-#1: 5′-GCGAUGUGAAUAGAGACUA-3′.
si-MTMR7-#2: 5′-CGGCCUAAACUUAAUGCAA-3′.
si-FLNB-#1: 5′-GCACGGUCACUGUUAGAUA-3′.

si-FLNB-#2: 5′-GAUCGUGUGAUGUCAAAUA-3′.

si-NC: 5′-UUCUCCGAACGUGUCACGU-3′.

The knockdown efficiency of all siRNAs was validated by qPCR prior to use. Transfection efficiency for both overexpression plasmids and siRNAs was further confirmed by Western blot analysis. The primers used for qPCR are listed below:

MTMR7-F: 5′-GTCCGCTTGGTAGATCGAGT-3′.

MTMR7-R: 5′-GTAGCGGTTGTTGCCTGTTT-3′.

FLNB-F: 5′-TTAAAGGTGACCCGAAGGGTG-3′.

FLNB-R: 5′-TGAAGGGACTGCGAGGAATC-3′.

18sRNA-F:5′-AAACGGCTACCACATCCAAG-3′.

18sRNA-R: 5′-CCTCCAATGGATCCTCGTTA-3′.

## Western blot assays

Protein samples were obtained through the lysis of cells, a process performed using the RIPA lysis buffer (Beyotime) as previously described [38]. The proteins were then denatured by heating for 10 minutes in a 100°C metal bath. Following electrophoretic separation, the proteins were transferred to PVDF membrane using a rapid wet transfer system (eBlotTM L1, GenScript). After completing the transfer, non-specific binding sites were blocked with 5% skim milk (Blotting Grade, Beyotime). It was then incubated with the anti-MTMR7 (1:1000, Proteintech), anti-FLNB (1:1000, Proteintech), anti-Ub (1:1000, Santa Cruz Blotechnology), anti-Tubulin (1:3000, Beyotime), and anti-Flag (1:1000, Sigma) primary antibodies at 4°C overnight. Samples were then incubated for 1 h with HRP-labeled goat anti-mouse/rabbit IgG secondary antibodies (Beyotime). At last, the protein signals were detected through the BeyoECL Plus/Moon kit (Beyotime) and analyzed with Image-Pro Plus software.

## Cell proliferation assays

In the CCK-8 assays, SSCs were inoculated in 96-well plates 48 h after transfection. Cell viability of SSCs was assessed through the Cell Counting Kit-8 (Beyotime) and the absorbance value was calculated at 450 nm with a microplate reader (Model 680, Bio-Rad) as previously described [39,40].

In the colony formation assays, cells that had been transfected for 48 h were seeded at equal densities into 6-well plates. After one week, the culture medium was refreshed. After 10–14 days, the culture medium was discarded, and methanol was then used to fix the cells for 20 minutes. The fixed cells were stained with crystal violet (Beyotime) and then analyzed for colony counting.

## Immunoprecipitation (IP)

Immunoprecipitation experiments were conducted using SSCs transfected with either Flag-MTMR7 or a negative control plasmid [41]. Following culture and harvesting, cells were lysed, and protein A/G magnetic beads (Vazyme) were added to the protein extracts to eliminate non-specific proteins. Subsequently, anti-Flag magnetic beads (Sigma) were introduced and the mixture was incubated overnight at 4°C on a shaker. The magnetic beads were collected the following day, washed with PBS and RIPA respectively. The final elution is done by heating at 100°C with SDS buffer. As previously described, liquid chromatography-tandem mass spectrometry (LC-MS/MS) was used to analyze the interacting proteins in the immunoprecipitation samples [40,42].

## Cell migration assays

Transwell assays were performed according to previous methods [43,44]. In brief, serum-free DMEM/F12 with transfected SSCs was added to the upper compartment of chambers (Corning, USA) with 8 μm pore membranes. The lower

compartment was then filled with DMEM/F12 complete medium. After 48 h, the cells were fixed and stained with methanol and crystal violet, respectively. Following staining, the transwell chambers were photographed using a microscope, and the cells were counted.

### Cycloheximide (CHX) assays

After transfection, SSCs were exposed to Cycloheximide (100 μg/mL) and incubated for 0, 4, 8, and 10 h. Subsequently, expression level of FLNB was assessed through Western blotting.

### Ubiquitination assays

48 h post-transfection, SSCs were exposed to MG132, a proteasome inhibitor, for 6 h, then lysed with RIPA buffer supplemented with 1% PMSF. The resulting cell lysates were incubated overnight with an anti-FLNB antibody. Following this, the protein A/G magnetic beads were added to the mixture and incubated at 4°C for 4 h. The precipitates were then eluted by heating with SDS buffer, and the proteins were analyzed via immunoblotting using an anti-Ub antibody.

### Immunofluorescence

Staining with immunofluorescence was conducted as previously described with minor modifications [45–47]. Samples were blocked with 1% (w/v) BSA and then incubated with anti-MTMR7 (1:200, Proteintech), anti-β-catenin (1:200, Cell Signaling Technology), and anti-PLZF (1:500, R&D) primary antibody at 4°C for 12h. After washing with PBS, the samples were incubated in the dark with Alexa-Fluor secondary antibody (Thermo Scientific, USA) for 1h to perform fluorescence stainin. Photographing with a confocal microscope (LSM800, Zeiss), then processed and analyzed with Zeiss Zen lite 3.9 software.

### Statistical analyses

Unpaired Student's t-test was used to assess the differences between two groups, while one-way ANOVA was employed for comparisons across three or more groups. P-value below 0.05 was statistically significant.

## Results

### MTMR7 inhibits SSCs proliferation and migration in vitro

We initially examined the cellular localization of MTMR7 in human testes using immunostaining. The results demonstrated that MTMR7 co-localizes with promyelocytic leukemia zinc finger (PLZF), which is served as a known marker for SSCs, in normal human testicular tissue (Fig 1A). To explore how MTMR7 affects human SSCs biologically, we employed two separate siRNAs (si-MTMR7-#1 and si-MTMR7-#2) to suppress MTMR7 expression in a human SSC cell line, as confirmed by Western blotting (Fig 1B and 1C). CCK-8 assays revealed a significant enhancement in SSCs proliferation following the knockdown of MTMR7 (Fig 1D). Moreover, MTMR7-deficient SSCs exhibited enhanced colony formation ability (Fig 1E and 1F). Transwell assays revealed that MTMR7 knockdown notably promoted the migratory ability of SSCs (Fig 1G and 1H).

Next, we transfected Flag-MTMR7 plasmid and empty vector into human SSCs, respectively. Western blot confirmed significant upregulation of MTMR7 in Flag-MTMR7-transfected SSCs (Fig 2A). Subsequent CCK-8 and colony formation assays demonstrated that overexpression of MTMR7 suppressed SSCs proliferation (Fig 2B–2D). Similarly, transwell assays showed that MTMR7 overexpression inhibited the migration of SSCs (Fig 2E and 2F).

In conclusion, our findings indicate that MTMR7 is present in human SSCs and acts to inhibit their proliferation and migration in vitro.

### MTMR7 interacts with FLNB in SSCs

To identify the proteins that interact with MTMR7 in human SSCs, we separately transfected Flag-MTMR7 plasmid and empty vector into SSCs. Proteins were extracted by lysing the transfected SSCs and immunoprecipitation (IP)

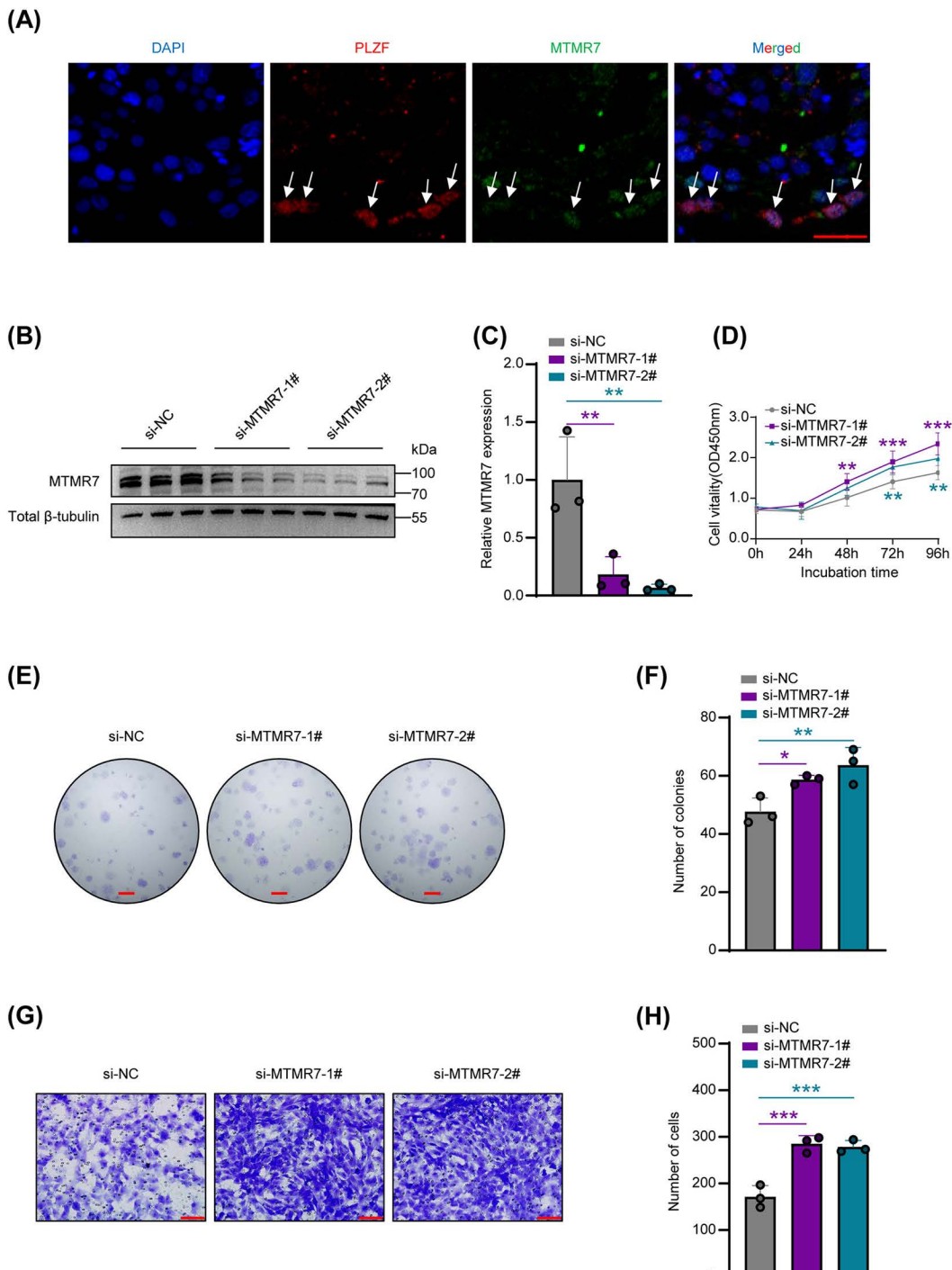

**Fig 1. MTMR7 knockdown promotes proliferation and migration of human SSCs.** (A) Co-immunostaining of PLZF and MTMR7 in adult human testis. Scale bar: 25 µm. PLZF served as a specific marker for human SSCs. (B) The efficiency of MTMR7 knockdown was validated using Western blot. (C) Quantitative analysis of the knockdown efficiency. n=3 per group. (D) Cell viability was measured using CCK-8 assays (n=6). (E) The proliferation of SSCs was evaluated using colony formation assays. Scale bar: 2 mm. (F) The results from (E) were quantified. n=3 per group. (G) Transwell assays of SSCs migration. Scale bar: 100 µm. (H) The quantification of migration data from (G) was performed. n=3 per group. *p<0.05, **p<0.01, ***p<0.001.

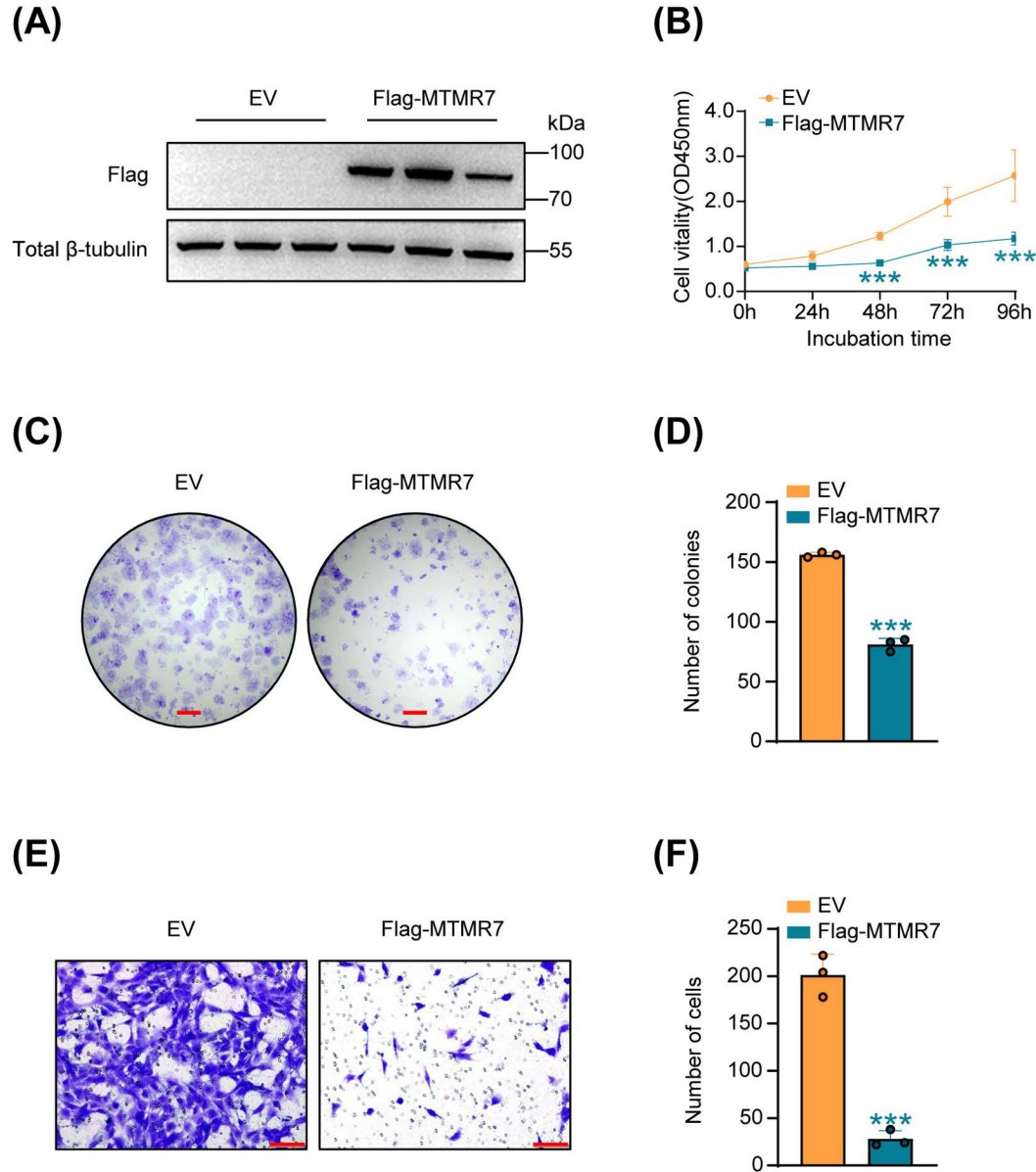

**Fig 2. Elevated levels of MTMR7 suppress the proliferation and migration of SSCs.** (A) Validation of MTMR7 overexpression in SSCs by Western blot. n = 3 per group. (B) CCK-8 assays of cell viability in SSCs (n = 6). (C) Results of colony-formation assays in SSCs. Scale bar: 2 mm. (D) The data from (C) were analyzed quantitatively. n = 3 per group. (E) Migration of SSCs transfected with Flag-MTMR7 plasmid or empty vector assessed by Transwell assays. Scale bar: 100 μm. (F) The migration results from (E) were quantified. n = 3 per group. ***p < 0.001.

experiments were performed using Flag-beads (to immunoprecipitate Flag-MTMR7). Subsequently, the immunoprecipitated proteins underwent PAGE, gel digestion, and LC-MS/MS (Fig 3A). Partial results of the LC-MS/MS analysis are presented in Fig 3B, with the full data available in S1 Table. We selected the interacting protein FLNB for further studies, as it ranked second only to MTMR7 itself in the number of unique peptides identified. Then, we validated the interaction between MTMR7 and FLNB in SSCs through reciprocal co-IP experiments (Fig 3C and 3D).

Furthermore, the interaction between MTMR7 and FLNB was explored through three-dimensional (3D) structural modeling and molecular docking analysis using HawkDock. The docking results were subsequently visualized with PyMOL. As shown in Fig 3E, the predicted 3D model illustrates the potential binding interaction and interface between MTMR7 and FLNB, with detailed binding sites provided in S2 Table. Collectively, these findings suggest a direct interaction between MTMR7 and FLNB.

### FLNB knockdown reduces SSCs proliferation and migration in vitro

After confirming the interaction between MTMR7 and FLNB, we further investigated the functional role of FLNB in SSCs by transfecting cells with two independent siRNAs (si-FLNB-1# and si-FLNB-2#) and confirmed the knockdown efficiency of FLNB via Western blotting (Fig 4A and 4B). FLNB knockdown led to a marked reduction in SSCs proliferation, as evidenced by CCK-8 and colony formation assays (Fig 4C–4E). Similarly, transwell assays indicated that the migratory ability of SSCs was reduced after FLNB knockdown (Fig 4F and 4G). Notably, these phenotypes closely resembled those observed upon MTMR7 overexpression, suggesting that FLNB may act as a key downstream effector of MTMR7 in regulating SSCs function. Based on these findings, we speculate that MTMR7 may influence SSCs proliferation and migration by modulating FLNB expression or activity.

### MTMR7 destabilizes FLNB protein and promotes its ubiquitin-mediated degradation

Western blot analysis revealed that MTMR7 overexpression markedly reduced FLNB protein levels in SSCs (Fig 5A and 5B), while MTMR7 knockdown led to a significant increase (Fig 5C and 5D). To further investigate the effect of MTMR7 on FLNB protein stability, we transfected SSCs with Flag-MTMR7 plasmid or empty vector and treated them with the protein synthesis inhibitor cycloheximide. The results showed that MTMR7 overexpression significantly reduced the half-life of endogenous FLNB in SSCs, rendering it unstable (Fig 5E and 5F). Moreover, we investigated the effect of MTMR7 on FLNB ubiquitination. Following transfection of SSCs with either Flag-MTMR7 plasmid or empty vector and treatment with the proteasome inhibitor MG132, IP experiments revealed that MTMR7 overexpression significantly increased the poly-ubiquitination of FLNB (Fig 5G).

These findings collectively indicate that MTMR7 downregulates FLNB protein levels in SSCs by enhancing its ubiquitination and promoting proteasomal degradation.

### The interaction between MTMR7 and FLNB reduces the expression of β-catenin

As a multifunctional protein, β-catenin serves as a key signaling molecule in the Wnt signaling pathway, regulating gene expression and participating in various biological processes [48]. Previous studies have indicated that β-catenin acts as a downstream factor following FLNB mutation, ultimately leading to 46, XY gonadal dysgenesis [49]. To investigate the association between β-catenin, MTMR7, and FLNB, we performed immunofluorescence analysis. Our results demonstrated that β-catenin expression was significantly inhibited after overexpression of MTMR7, whereas β-catenin levels was enhanced following MTMR7 knockdown (Fig 6A and 6B). Furthermore, β-catenin expression was also decreased after FLNB knockdown (Fig 6C and 6D).

Collectively, our findings suggest that in human SSCs, MTMR7 interacts with FLNB to form a co-regulatory axis, which may regulate β-catenin expression and ultimately inhibits the proliferative and migratory capacities of SSCs (Fig 7).

## Discussion

Our previous study demonstrated that MTMR7 is specifically expressed in neonatal mouse SSCs and plays a regulatory role in their proliferation. The present findings extend this understanding to human SSCs, as we observed that MTMR7 is also expressed in human testes and partially co-localizes with the SSCs marker PLZF. Moreover, consistent with its previously reported anti-proliferative effects in various somatic cell types, MTMR7 was found to suppress both proliferation and migration of human SSCs in this study.

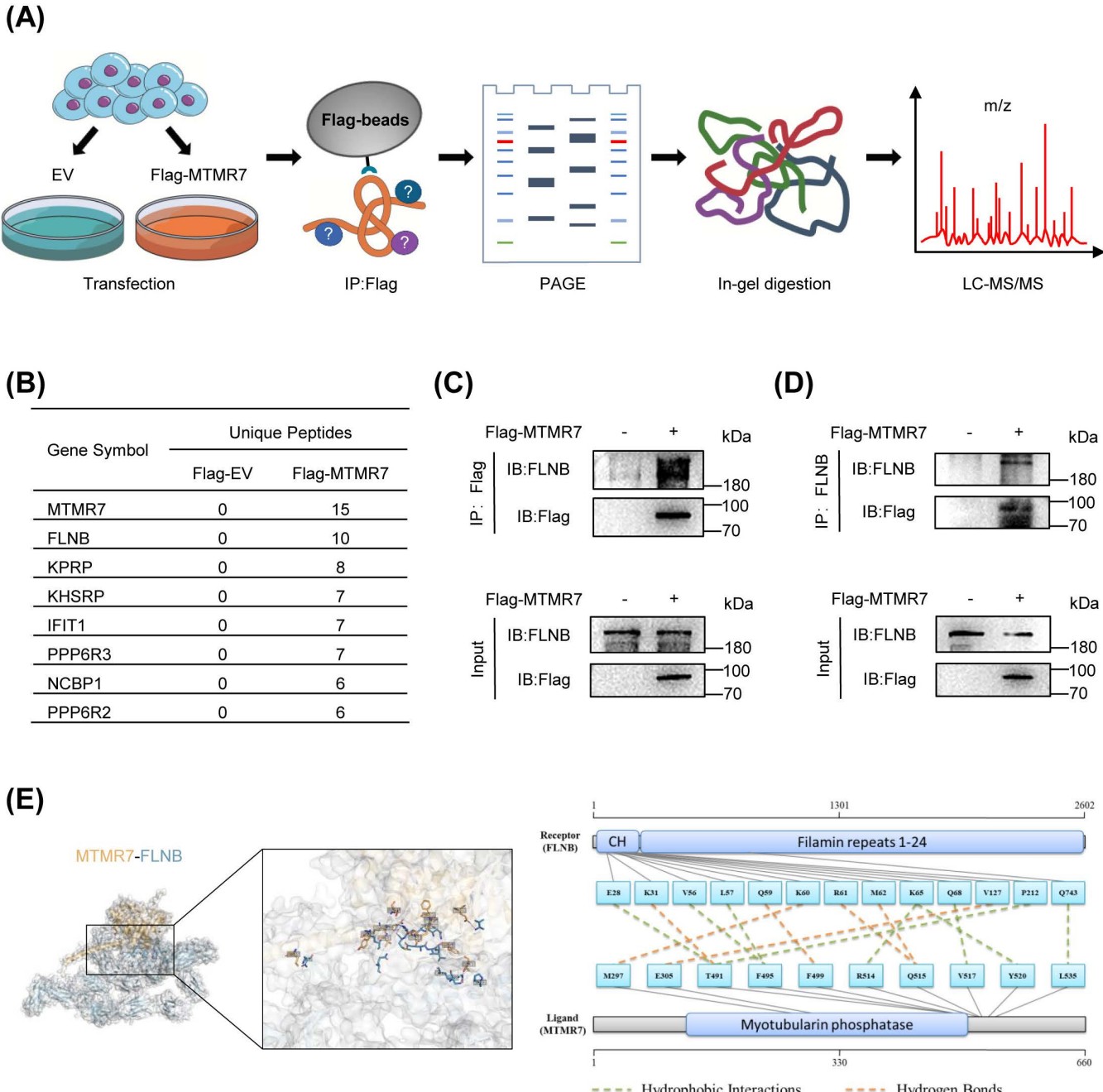

**Fig 3. MTMR7 interacts with FLNB in SSCs.** (A) Procedures for identifying proteins interacting with MTMR7 in SSCs: total protein was extracted after transfection of SSCs with Flag-MTMR7 plasmid or empty vector, and subjected to Flag-beads immunoprecipitation, PAGE, in-gel digestion, and LC-MC/MS analysis. (B) Results of LC-MC/MS analysis: interacting proteins ranked by the number of unique peptides identified. (C) SSCs were transfected with Flag-MTMR7 or an empty vector. Protein extracts were immunoprecipitated with Flag-beads or (D) anti-FLNB antibodies. (E) 3D structural model of the MTMR7 (orange) and FLNB (blue) complex, binding residues are indicated.

To further investigate the mechanism of MTMR7 in human SSCs, we identified FLNB as an interacting protein of MTMR7 through immunoprecipitation and LC-MS/MS analysis. Our results showed that overexpression of MTMR7

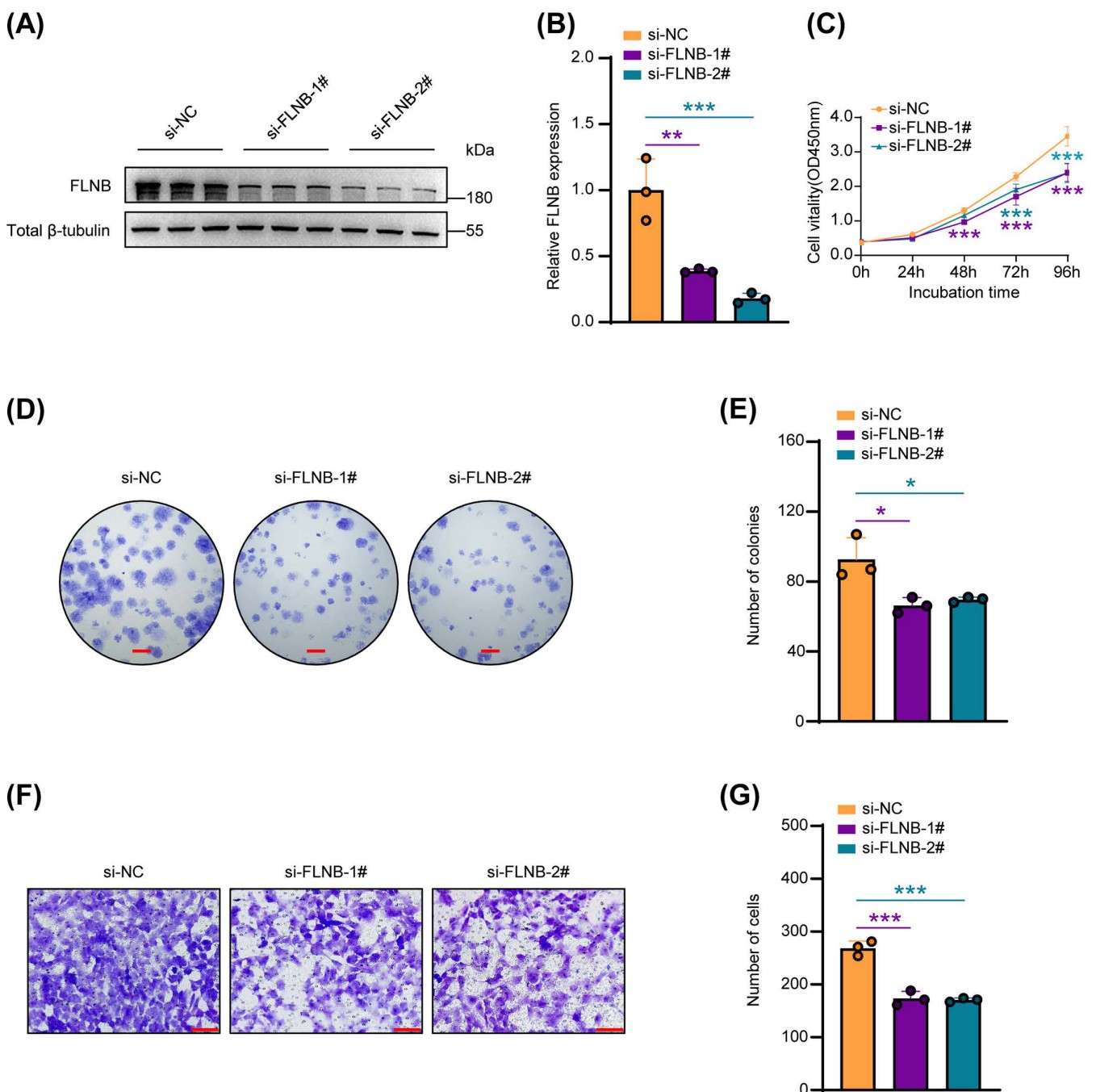

**Fig 4. Impact of FLNB knockdown on SSCs proliferation and migration.** (A) Western blot was used to examine FLNB knockdown in SSCs. (B) The results from (A) were quantified. n = 3 per group. (C) Cell viability was evaluated using CCK-8 assays (n = 6). (D) SSCs proliferation was assessed through colony formation assays. Scale bar: 2 mm. (E) Quantitative assessment of colony formation from (D). n = 3 per group. (F) Evaluation of SSCs migration using Transwell assays. Scale bar: 100 μm. (G) Quantification of (F). n = 3 per group. *p < 0.05, **p < 0.01, ***p < 0.001.

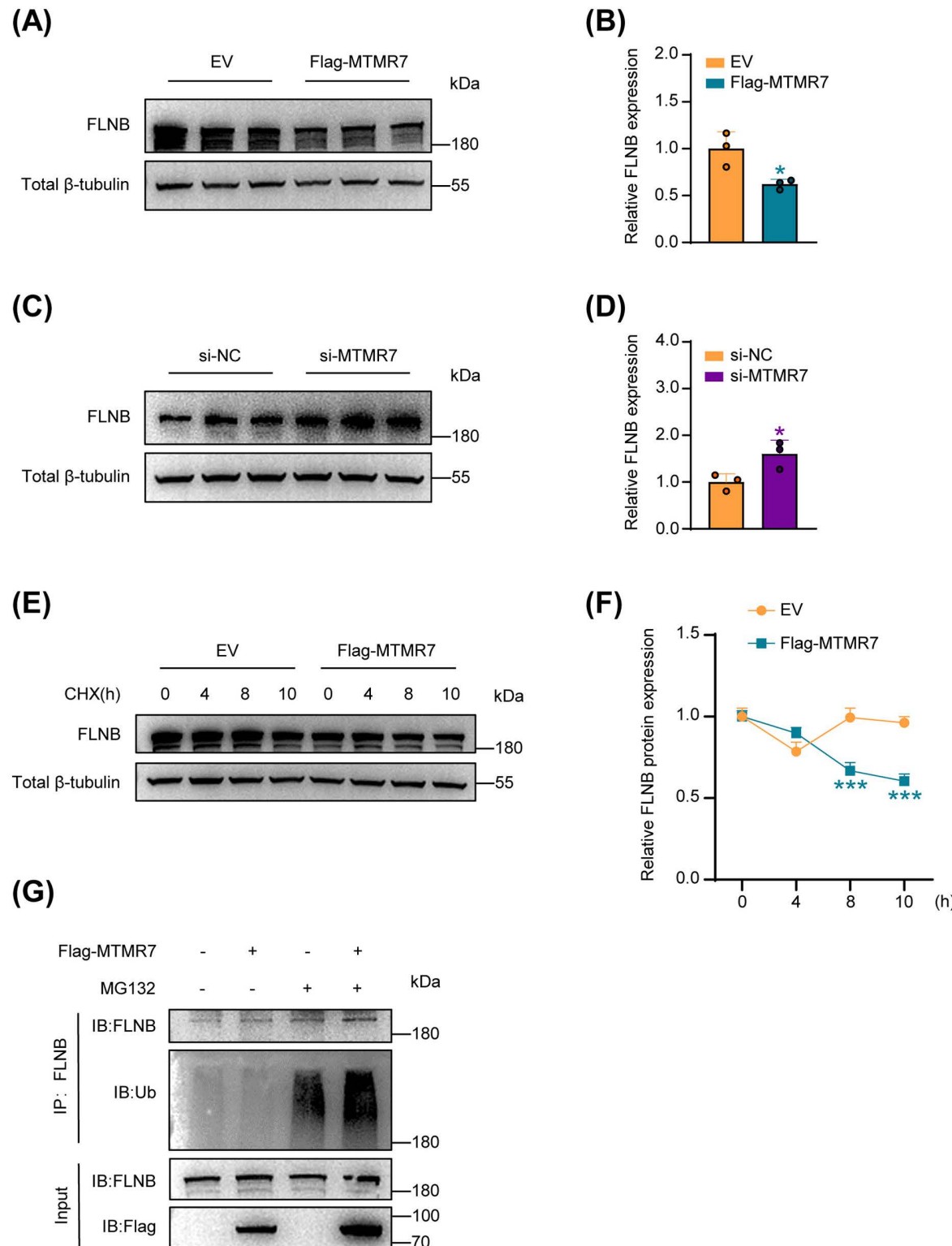

**Fig 5. MTMR7 promotes the ubiquitin-mediated degradation of FLNB.** (A) Expression levels of FLNB in SSCs transfected with Flag-MTMR7 plasmid or empty vector. (B) results from (A) were quantitatively analyzed. n = 3 per group. (C) FLNB expression levels in SSCs with knockdown of MTMR7

assessed using western blotting. (D) quantitative analysis of the data from (C). n = 3 per group. (E) SSCs were transfected with Flag-MTMR7 plasmid or empty vector and then treated with 100 μM cycloheximide for 0, 4, 8, and 10 h. Subsequent Western blot analysis showed that overexpression of MTMR7 decreased the half-life of FLNB. (F) Quantification of (E), with results replicated across three independent experiments. (G) After transfection with Flag-MTMR7 plasmid or empty vector, SSCs were treated with MG132 and then immunoprecipitated using an anti-FLNB antibody. The ubiquitination levels of FLNB was detected by Western blotting using an anti-Ub antibody. Results were consistent across three separate experiments. *p < 0.05, ***p < 0.001.

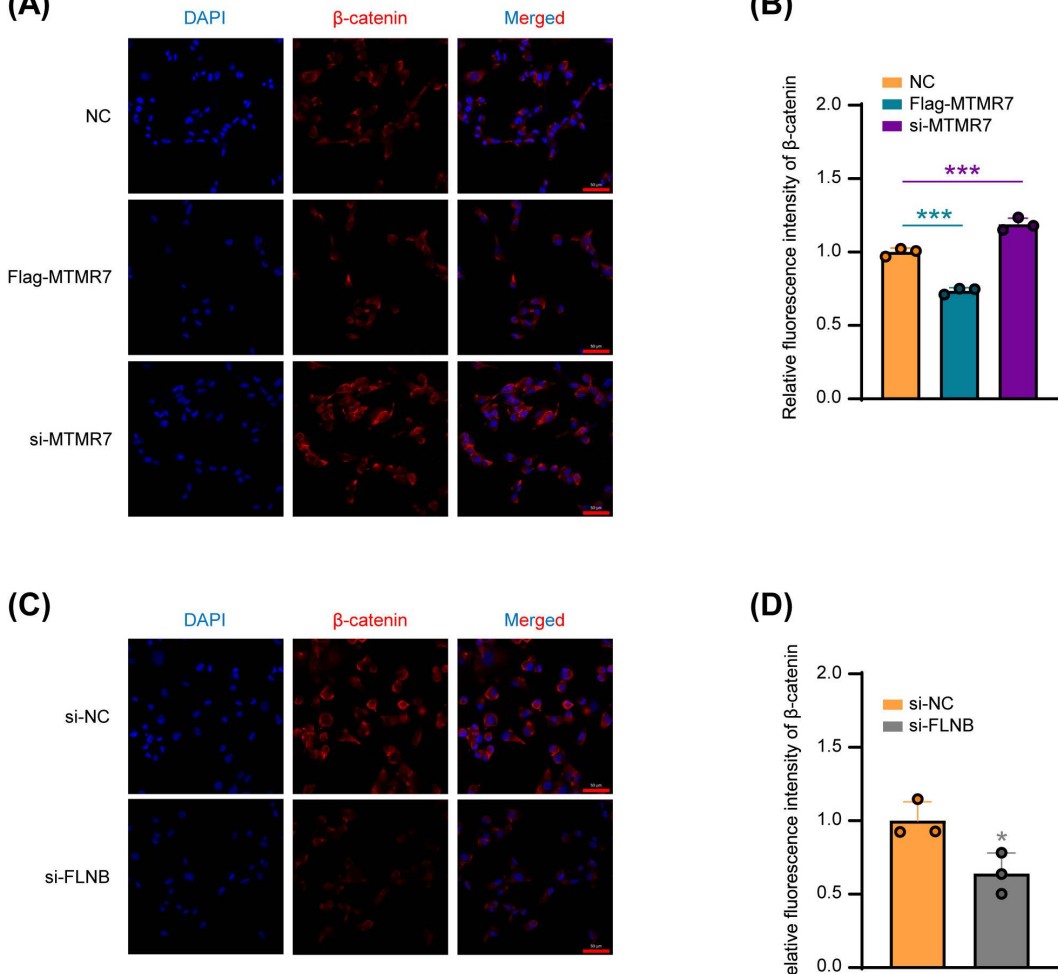

**Fig 6. MTMR7–FLNB interaction suppresses. β-catenin expression.** (A) Immunofluorescence detection of β-catenin expression in SSCs following transfection with NC, si-MTMR7 or Flag-MTMR7 plasmid. (B) Quantitative analysis of β-catenin expression from (A). n = 3 per group. (C) β-Catenin expression in SSCs detected by immunofluorescence after transfection with si-NC or si-FLNB. (D) Quantitative assessment of β-catenin levels from (C). n = 3 per group. *p < 0.05, ***p < 0.001.

significantly downregulated FLNB expression, while knockdown of MTMR7 increased FLNB expression levels. Additionally, we observed that the half-life of endogenous FLNB was markedly shortened with MTMR7 overexpression, because MTMR7 accelerated the degradation of FLNB through the ubiquitination pathway.

FLNB, a key actin-binding protein, plays an essential role in cytoskeletal organization and the regulation of cell motility. Previous studies have shown that FLNB knockdown in various cancer cell types—including ovarian, cervical,

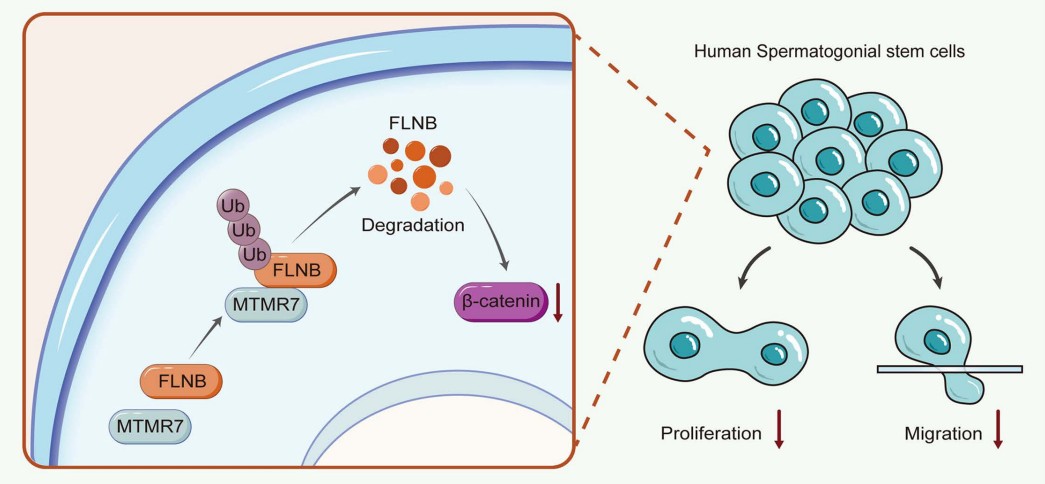

**Fig 7. Schematic model of MTMR7 function in human SSCs.** MTMR7 promotes the ubiquitination and degradation of the substrate protein FLNB, which subsequently inhibits the expression levels of β-catenin, ultimately affecting the proliferation and migration of SSCs.

and renal clear cell carcinomas—enhances their proliferative capacity and invasiveness [50–52]. In contrast, deletion of FLNB in chondrocytes impairs cell proliferation and leads to premature differentiation [53,54]. Additionally, FLNB functions as a scaffold protein linking VEGFR2, Vav-2, and Rac-1 in vascular endothelial cells, thereby regulating cell migration and promoting angiogenesis; its deficiency results in impaired endothelial cell migration [55]. Downregulation of FLNB also contributes to decreased expression of ERK, MMP-2, and MMP-9, which in turn reduces placental trophoblast invasion in preeclampsia [56]. In the context of male reproduction, FLNB mutations have been found to enhance the interaction with RAC1 and MAP3K1, thereby activating β-catenin signaling and contributing to 46, XY gonadal dysgenesis [49]. In the present study, we found that FLNB knockdown significantly suppressed both the proliferation and migration of human SSCs. These results highlight a potential dual and context-dependent role of FLNB—suppressing cell proliferation and invasion in malignant cells, while potentially promoting these processes in normal physiological contexts. Further studies are required to elucidate the molecular mechanisms underlying this context specificity.

As a pivotal effector of the canonical Wnt pathway, β-catenin plays a critical regulatory role in cell proliferation, differentiation, and development. β-Catenin is phosphorylated and targeted for degradation through a multiprotein complex consisting of APC, Axin, GSK-3, and CK1 in the absence of Wnt signaling. Conversely, if Wnt signaling is present, the complex is inhibited, leading to β-catenin accumulation in the cytoplasm. This β-catenin then moves to the nucleus to activate target gene expression [57,58].

The function of β-catenin in mouse SSCs is still debated. Some research indicates that high levels of β-catenin favor the growth and development of SSCs. For example, the Eif2s3y gene has been shown to promote SSCs proliferation through the activation of the Wnt6/β-catenin signaling pathway [59]. Wnt3A and Wnt10B enhance SSCs proliferation by stabilizing and accumulating cytoplasmic β-catenin [60]. Moreover, paracrine signaling through the Wnt/β-catenin pathway has been suggested to play a role in regulating SSC proliferation [61]. However, other studies have indicated that aberrant activation of the Wnt/β-catenin pathway can be detrimental. Knockdown of CNBP in neonatal testicular cultures results in seminiferous tubule disruption, Sertoli cell mislocalization, and germ cell loss, which is associated with abnormal activation of the Wnt/β-catenin pathway [62]. Furthermore, BMI1 has been shown to maintain SSC proliferation through the epigenetic inhibition of Wnt10b/β-catenin signaling [45].

In human SSCs, β-catenin also plays a critical role. Downregulation of HOXC5 leads to a significant reduction in nuclear β-catenin levels, ultimately inhibiting SSCs proliferation and promoting apoptosis [63]. Conversely, FLNB mutations have been shown to activate β-catenin, resulting in 46, XY gonadal dysgenesis [49]. In our study, we found that overexpression of MTMR7 and knockdown of FLNB decreased β-catenin levels, thereby inhibiting human SSCs proliferation. In contrast, knockdown of MTMR7 increased β-catenin levels, promoting human SSCs proliferation. These findings highlight the context-dependent nature of β-catenin signaling in SSCs regulation.

Downstream of β-catenin, target genes such as TCF1, Cyclin D1, Axin2, Lgr5, and c-Myc promote cell cycle progression in both tumor cells and SSCs [63–66]. MMP7, another β-catenin target implicated in hepatocellular carcinoma cell migration [67], may also contribute to SSCs motility. In our study, MTMR7-FLNB interaction led to reduced β-catenin levels, which may suppress SSCs proliferation via these downstream effectors. Although their expression was not directly examined, future studies exploring their roles within the MTMR7/FLNB/β-catenin axis may shed light on the molecular mechanisms governing SSCs fate.

Despite uncovering the biological significance of the MTMR7/FLNB/β-catenin axis in SSCs, this study has several limitations. First, the use of a single human SSC line (hSSC-1) may not fully represent the heterogeneity of primary SSCs. Second, the absence of in vivo validation limits our understanding of this axis in a physiological context. From a clinical perspective, MTMR7 and FLNB may serve as potential diagnostic and therapeutic targets for NOA, particularly in patients with impaired SSCs proliferation. However, several challenges remain, including achieving testis-specific delivery while minimizing systemic effects, and balancing SSCs proliferation and differentiation.

## Conclusion

Collectively, in human SSCs, MTMR7 enhances the ubiquitin-mediated degradation of FLNB. This interaction subsequently suppresses the β-catenin signaling pathway, resulting in reduced proliferation and migration of cells. Our findings elucidate novel regulatory mechanisms governing human SSCs fate and highlight potential therapeutic targets for NOA associated with SSCs dysfunction.

## Supporting information

**S1 Table. Full list of MTMR7-interacting proteins identified by LC-MS/MS.**
(XLSX)

**S2 Table. List of interaction pairs between FLNB and MTMR7.**
(DOCX)

**S1 raw images. Original uncropped Western blot images corresponding to main text figures.**
(PDF)

## Author contributions

**Data curation:** Nianchao Zhou.

**Formal analysis:** Yi Yu, Bing Jiang, Haoyue Hu, Xiaoyan Huang.

**Methodology:** Tingting Gao.

**Resources:** Wenxin Gao.

**Validation:** Nianchao Zhou, Tiantian Wu.

**Writing – original draft:** Cong Shen.

**Writing – review & editing:** Yibo Wu.

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
