## [Decision Letter · Decision Letter 0]

PONE-D-24-58024MTMR7 regulates human SSCs proliferation and migration via targeting FLNBPLOS ONE

Dear Dr. Shen,

Thank you for submitting your manuscript to PLOS ONE. After careful consideration, we feel that it has merit but does not fully meet PLOS ONE’s publication criteria as it currently stands. Therefore, we invite you to submit a revised version of the manuscript that addresses the points raised during the review process.

We look forward to receiving your revised manuscript.

Kind regards,

Jeevithan Elango, PhD

Academic Editor

PLOS ONE

“This work was supported by the National Natural Science Foundation of China 420 (82201762), the Suzhou Gu Su Health Talent Research Project (GSWS2023056), and 421 the Top Talent Support Program for Young and Middle-Aged People of Wuxi Health 422 Committee (BJ2020047).”

“This work was supported by the National Natural Science Foundation of China (82201762), the Suzhou Gu Su Health Talent Research Project (GSWS2023056), and the Top Talent Support Program for Young and Middle-Aged People of Wuxi Health Committee (BJ2020047).”

“This work was supported by the National Natural Science Foundation of China 420 (82201762), the Suzhou Gu Su Health Talent Research Project (GSWS2023056), and 421 the Top Talent Support Program for Young and Middle-Aged People of Wuxi Health 422 Committee (BJ2020047).”

4. We note that Figure 6 in your submission contain copyrighted images. All PLOS content is published under the Creative Commons Attribution License (CC BY 4.0), which means that the manuscript, images, and Supporting Information files will be freely available online, and any third party is permitted to access, download, copy, distribute, and use these materials in any way, even commercially, with proper attribution. For more information, see our copyright guidelines: http://journals.plos.org/plosone/s/licenses-and-copyright.

1. You may seek permission from the original copyright holder of Figure 6 to publish the content specifically under the CC BY 4.0 license.

Reviewers' comments:

Reviewer's Responses to Questions

**Comments to the Author**

1. Is the manuscript technically sound, and do the data support the conclusions?

Reviewer #1: Yes

Reviewer #2: Partly

Reviewer #3: Yes

2. Has the statistical analysis been performed appropriately and rigorously? 

Reviewer #1: Yes

Reviewer #2: Yes

Reviewer #3: Yes

3. Have the authors made all data underlying the findings in their manuscript fully available?

Reviewer #1: Yes

Reviewer #2: Yes

Reviewer #3: Yes

4. Is the manuscript presented in an intelligible fashion and written in standard English?

Reviewer #1: Yes

Reviewer #2: Yes

Reviewer #3: Yes

5. Review Comments to the Author

Reviewer #1: The study is promising and provides valuable insights into the role of MTMR7 in human SSCs. However, major revisions are needed to address the points above, particularly in terms of additional validation, clarification of methods, and discussion of limitations.

1. Introduction:

o Clarify the rationale for focusing on MTMR7 in human SSCs, especially compared to other MTMR family members. Expand on why MTMR7 is particularly relevant in the context of human SSCs.

2. Materials and Methods:

o Provide more details on the donor's clinical background (age, health status) for the human testicular tissues.

o Include additional characterization of the human SSC line (e.g., markers used to confirm its identity as SSCs).

o Add data on siRNA efficiency and potential off-target effects for MTMR7 and FLNB knockdown experiments.

3. Results:

o Validate the MTMR7-FLNB interaction further using additional techniques like co-localization studies or more Co-IP experiments.

o Include controls in ubiquitination assays to confirm specificity of FLNB ubiquitination.

o Explore downstream effects of β-catenin on SSCs proliferation and migration, such as identifying specific Wnt/β-catenin target genes.

4. Discussion:

o Address limitations of the study, such as the reliance on a single human SSC line and the lack of in vivo validation.

o Discuss clinical implications of targeting MTMR7 or FLNB for male infertility treatment, including potential drugs or challenges in translation.

5. Figures and Data Presentation:

o Improve labeling in graphs (e.g., Fig 1D, 1E, 2B, 2C) to make time points and conditions clearer.

o Provide a detailed table of LC-MS/MS results in the supplementary materials, listing all identified interacting proteins.

6. Supplementary Materials:

o Include full Western blot images, additional validation experiments, and a more comprehensive table of LC-MS/MS results.

Reviewer #2: Overall Impression:

This manuscript presents an interesting study on the role of MTMR7 in regulating human spermatogonial stem cells (SSCs) proliferation and migration through its interaction with FLNB and subsequent effects on β-catenin signaling. The study provides novel insights into the molecular mechanisms underlying SSCs regulation and potential therapeutic targets for male infertility. However, there are several areas that require substantial revision to enhance the clarity, robustness, and impact of the findings.

From human testis single cell sequencing data, RNA sequencing data from SSC cells, SSC protein mass spectrometry, and data from the Human protein atlas database, it appears that MTMR7 is a gene with very low or even difficult to detect expression. Is it present in SSC and why did the authors bother to study its role? Please explain the reasons in detail.

1.Introduction:

The introduction provides a good background on the importance of SSCs in spermatogenesis and male fertility. However, it could benefit from a more detailed discussion on the differences between mouse and human SSCs, particularly regarding the molecular pathways involved. This would help contextualize the significance of studying MTMR7 in human SSCs.

Suggestion: Expand the discussion on the differences in molecular mechanisms between mouse and human SSCs to better justify the need for this study.

2.Materials and Methods:

The use of si-RNAs and overexpression plasmids is well described, but the authors should provide more information on the efficiency and specificity of these tools. For example, were off-target effects of si-RNAs assessed?

Suggestion: Clearly specify the sample size for each experiment and provide additional details on the efficiency and specificity of the si-RNAs and overexpression plasmids used.

3.The data on FLNB knockdown and its effects on SSCs proliferation and migration are presented, but the connection to MTMR7's role is not fully explored. The authors should provide a more detailed analysis of how FLNB knockdown mimics the effects of MTMR7 overexpression.

Suggestion: Enhance the clarity of the results by providing a more detailed analysis of the relationship between MTMR7, FLNB, and β-catenin signaling. Consider including additional experiments to further elucidate this relationship.

4.the immunofluorescence images in Figure 5 could be enhanced with more detailed annotations.

5.

This manuscript presents a valuable study on the role of MTMR7 in regulating human SSCs proliferation and migration. However, it requires substantial revision to address the issues outlined above.

Recommended Revision: Major Revision

Sincerely,

Dai Zhou

Reviewer #3: This research is well-designed, with a clear hypothesis, and highlights the function of MTMR7 in hSSCs, offering potential insights for the clinical understanding of male infertility. However, the manuscript requires improvements in language expression and structural organization. Detailed comments are provided below:

1. Introduction Section: The background information on SSCs is insufficient. It would be beneficial to include more details on the biological roles of SSCs and their regulatory mechanisms to provide a stronger foundation for the study.

2. Structural Organization: The background information on FLNB, currently included in the Introduction, should be moved to the Discussion section. Since FLNB was identified through subsequent IP experiments, this adjustment would improve the logical flow of the manuscript.

Language Simplification:

3. Some sentences are overly verbose and can be streamline:

Line 224: “Fig 1. Reducing MTMR7 levels enhances both the proliferation and migration of human SSCs. ” could be revised to “Fig 1. MTMR7 knockdown promotes proliferation and migration of human SSCs. ”

Line 235: “Western blot analysis confirmed substantial upregulation of MTMR7 protein expression in SSCs transfected with the Flag-MTMR7 plasmid” could be simplified to “Western blot confirmed significant upregulation of MTMR7 in Flag-MTMR7-transfected SSCs. ”

4. Formal Academic Tone: Some phrases are too informal and should be revised to align with academic writing standards. For example:

Line 416: “These results provide new insights into the regulatory mechanisms affecting human SSCs fate and suggest potential new targets for treating NOA linked to abnormal SSCs development.” could be rephrased as “Our findings elucidate novel regulatory mechanisms governing human SSCs fate and highlight potential therapeutic targets for NOA associated with SSCs dysfunction. ”

5. Reference Formatting: Several references are incomplete or incorrectly formatted.

Line 461: “8. <pnas-0407063101.pdf>.”

Line 520: “24. <10.1007@978-3-030-24108-7.pdf>.”

These should be revised to include complete citation details in the correct format.</pnas-0407063101.pdf>

6. PLOS authors have the option to publish the peer review history of their article (what does this mean? ). If published, this will include your full peer review and any attached files.

**Do you want your identity to be public for this peer review?** For information about this choice, including consent withdrawal, please see our Privacy Policy .

Reviewer #1: **Yes: ** Alaa Sayed Abou-Elhamd

Reviewer #2: **Yes: ** Dai Zhou

Reviewer #3: No

---

## [Author Response · Author response to Decision Letter 1]

7 Jun 2025

Detailed Responses to Reviewer Comments

Reviewer 1:

1. Introduction:

o Clarify the rationale for focusing on MTMR7 in human SSCs, especially compared to other MTMR family members. Expand on why MTMR7 is particularly relevant in the context of human SSCs.

Response: Thank you for your comment. We have revised the Introduction section to clarify the rationale for focusing on MTMR7 in human SSCs, as compared to other MTMR family members. The revised paragraph is as follows:

“As a key member of the MTMR family, myotubularin-related protein 7 (MTMR7) differs from other members that primarily dephosphorylate PtdIns3P and PtdIns(3,5)P₂, as it exhibits a stronger catalytic preference for its specific substrate, inositol 1,3-bisphosphate (Ins(1,3)P₂) [25]. This unique enzymatic activity may underlie its distinct biological functions. MTMR7 has been shown to significantly inhibit proliferation across a range of cell types, such as gastric cancer cells, vascular smooth muscle cells, colorectal cancer cells and myoblasts [26-29]. In the context of male reproduction, our previous studies demonstrated that MTMR7 is specifically expressed in SSCs in neonatal mouse testes and contributes to the maintenance of cell cycle homeostasis by negatively regulating the PI3K/AKT signaling pathway [30]. These findings suggest that MTMR7 may also serve as a potential biomarker in human SSCs. However, given the phenotypic and regulatory differences between human and mouse SSCs, the precise role and underlying molecular mechanisms of MTMR7 in human SSCs remain to be elucidated.”

2. Materials and Methods:

o Provide more details on the donor's clinical background (age, health status) for the human testicular tissues.

Response: Thank you for this suggestion. We have revised the Sample collection section to provide more details on the donor's clinical background. The revised paragraph is as follows:

“Human testicular tissues were obtained from a 34-year-old healthy male donor with normal spermatogenesis, whose partner was receiving conventional in vitro fertilization (IVF) treatment at Suzhou Municipal Hospital. The tissue samples were collected during the recruitment period from 30 October 2023 to 1 November 2023. Paraffin sections were prepared by rinsing tissues with phosphate-buffered saline (PBS) and fixing in 4% paraformaldehyde [36, 37]. This study was approved by the Ethics Committee of Suzhou Municipal Hospital (No.2023005) and a written informed consent from the donor was obtained in accordance with the Declaration of Helsinki.”

o Include additional characterization of the human SSC line (e.g., markers used to confirm its identity as SSCs).

Response: Thanks again for your suggestion. We have revised the Cell culture section to include additional characterization of the human SSC line. The revised paragraph is as follows:

“The human SSC line was generously provided by Professor Zuping He (Hunan Normal University, China). These cells express characteristic SSCs markers, including GPR125, GFRA1, and PLZF, as confirmed by immunocytochemistry [38, 39]. The cells were cultured in DMEM/F-12 medium (Gibco) to support their growth, with additional supplements including 10% fetal bovine serum (FBS, ScienCell, USA) to provide essential nutrients, and 1% penicillin-streptomycin solution to prevent bacterial contamination.”

o Add data on siRNA efficiency and potential off-target effects for MTMR7 and FLNB knockdown experiments.

Response: Based on your suggestion, we have revised the section on si-RNA, plasmids, and transfection reagents by incorporating content addressing siRNA efficiency and potential off-target effects. The updated section now reads as follows:

“When the cell density reached approximately 70%, overexpression plasmids pcDNA3.1-Flag-MTMR7 and an empty vector (Sangon, Shanghai, China) were introduced into SSCs using the X-treme GENE HP DNA transfection reagent (Mannheim, Germany). For plasmid transfections, 10 μg DNA was used per 10-cm dish and 2 μg per well of a 6-well plate. Separately, siRNAs targeting MTMR7 and FLNB (GenePharma, Suzhou, China) were transfected into SSCs using Lipofectamine 2000 (Invitrogen, USA), with 10 μl siRNAs per well in a 6-well plate [40-42]. To minimize off-target effects, two independent siRNAs were designed for each gene, and only consistent results from both were considered valid. The sequences targeted by the siRNAs were as follows:

si-MTMR7-#1: 5′-GCGAUGUGAAUAGAGACUA-3′.

si-MTMR7-#2: 5′-CGGCCUAAACUUAAUGCAA-3′.

si-FLNB-#1: 5′-GCACGGUCACUGUUAGAUA-3′.

si-FLNB-#2: 5′-GAUCGUGUGAUGUCAAAUA-3′.

si-NC: 5′-UUCUCCGAACGUGUCACGU-3′.

The knockdown efficiency of all siRNAs was validated by qPCR prior to use. Transfection efficiency for both overexpression plasmids and siRNAs was further confirmed by Western blot analysis. The primers used for qPCR are listed below:

MTMR7-F: 5′-GTCCGCTTGGTAGATCGAGT-3′.

MTMR7-R: 5′-GTAGCGGTTGTTGCCTGTTT-3′.

FLNB-F: 5′-TTAAAGGTGACCCGAAGGGTG-3′.

FLNB-R: 5′-TGAAGGGACTGCGAGGAATC-3′.

18sRNA-F:5′-AAACGGCTACCACATCCAAG-3′.

18sRNA-R: 5′-CCTCCAATGGATCCTCGTTA-3′.”

3. Results:

o Validate the MTMR7-FLNB interaction further using additional techniques like co-localization studies or more Co-IP experiments.

Response: Thank you for the valuable suggestion. We have further validated the interaction between MTMR7 and FLNB by performing three-dimensional (3D) structural modeling and molecular docking analysis using HawkDock. The docking results were visualized with PyMOL. As presented in the revised manuscript (Fig. 3E) and detailed in S2 Table, the 3D model illustrates the potential binding interaction and interface between MTMR7 and FLNB. These findings provide additional support for a direct interaction between MTMR7 and FLNB.

o Include controls in ubiquitination assays to confirm specificity of FLNB ubiquitination.

Response: Thank you very much for this valuable comment. We fully agree that including appropriate controls would further enhance the rigor of our ubiquitination assay. In our current experimental design, we did include certain controls: the EV (empty vector) group was used to exclude non-specific effects of the vector itself, and MG132 treatment confirmed that FLNB ubiquitination is proteasome-dependent. The observed increase in FLNB ubiquitination upon MG132 treatment in the MTMR7 overexpression group supports the specificity of this modification. However, we acknowledge that additional controls such as IP with an unrelated IgG would further strengthen the conclusion. We sincerely appreciate your suggestion and will incorporate such additional controls in future studies to further validate the specificity of target protein ubiquitination.

o Explore downstream effects of β-catenin on SSCs proliferation and migration, such as identifying specific Wnt/β-catenin target genes.

Response: We sincerely appreciate your valuable suggestion regarding the exploration of downstream β-catenin target genes. Due to experimental constraints, we were unable to directly investigate these targets in the current study. However, in response to this helpful comment, we have expanded the Discussion section to include relevant literature and potential downstream effectors. We also highlighted the need for future studies to further elucidate their roles within the MTMR7/FLNB/β-catenin regulatory axis. The revised paragraph is as follows:

“Downstream of β-catenin, target genes such as TCF1, Cyclin D1, Axin2, Lgr5, and c-Myc promote cell cycle progression in both tumor cells and SSCs [65-68]. MMP7, another β-catenin target implicated in hepatocellular carcinoma cell migration [69], may also contribute to SSC motility. In our study, MTMR7-FLNB interaction led to reduced β-catenin levels, which may suppress SSCs proliferation via these downstream effectors. Although their expression was not directly examined, future studies exploring their roles within the MTMR7/FLNB/β-catenin axis may shed light on the molecular mechanisms governing SSCs fate.”

4. Discussion:

o Address limitations of the study, such as the reliance on a single human SSC line and the lack of in vivo validation.

Response: Thank you for this suggestion. In the revised Discussion, we have added statements addressing the limitations of using a single human SSC line and the lack of in vivo validation. The revised paragraph is as follows:

“Despite uncovering the biological significance of the MTMR7/FLNB/β-catenin axis in SSCs, this study has several limitations. First, the use of a single human SSC line (hSSC-1) may not fully represent the heterogeneity of primary SSCs. Second, the absence of in vivo validation limits our understanding of this axis in a physiological context.”

o Discuss clinical implications of targeting MTMR7 or FLNB for male infertility treatment, including potential drugs or challenges in translation.

Response: Thank you for this comment. In the revised Discussion, we have incorporated a section discussing the potential clinical implications of targeting MTMR7 or FLNB, as well as the challenges in translating these findings into therapeutic strategies. The revised paragraph is as follows:

“From a clinical perspective, MTMR7 and FLNB may serve as potential diagnostic and therapeutic targets for NOA, particularly in patients with impaired SSCs proliferation. However, several challenges remain, including achieving testis-specific delivery while minimizing systemic effects, and balancing SSCs proliferation and differentiation.”

5. Figures and Data Presentation:

o Improve labeling in graphs (e.g., Fig 1D, 1E, 2B, 2C) to make time points and conditions clearer.

Response: Thank you for this suggestion. In the revised manuscript, we have updated the labeling of the CCK-8 assay results (Fig 1D, 2B, 4C) to make the time points clearer and easier to interpret. Additionally, we have added scale bars to the colony formation assay images (Fig 1E, 2C, 4D) to improve clarity and consistency in data presentation. We appreciate your advice, which helped us improve the overall quality of our figures.

o Provide a detailed table of LC-MS/MS results in the supplementary materials, listing all identified interacting proteins.

Response: Thank you for this comment. We have provided a detailed table listing all identified MTMR7-interacting proteins from the LC-MS/MS analysis in the supplementary materials (see S1_Table).

6. Supplementary Materials:

o Include full Western blot images, additional validation experiments, and a more comprehensive table of LC-MS/MS results.

Response: We have included the full original Western blot images (S1_raw_images) and the complete LC-MS/MS results (S1_Table) in the supplementary materials as requested.

Reviewer 2:

From human testis single cell sequencing data, RNA sequencing data from SSC cells, SSC protein mass spectrometry, and data from the Human protein atlas database, it appears that MTMR7 is a gene with very low or even difficult to detect expression. Is it present in SSC and why did the authors bother to study its role? Please explain the reasons in detail.

Response:

Thank you very much for this important and insightful question. We fully acknowledge that MTMR7 expression appears to be low or even difficult to detect in public datasets such as single-cell RNA-seq or proteomic databases. However, this is in part due to the extreme rarity of SSCs within the human testis, making robust detection of SSC-specific transcripts or proteins inherently challenging in bulk or even single-cell analyses.

In our previous work, we identified MTMR7 in the proteomic profile of neonatal mouse testes and demonstrated that MTMR7 regulates SSCs cell cycle homeostasis via the PI3K/AKT signaling pathway. Building on these findings, we sought to investigate whether MTMR7 might also play a conserved role in human SSCs, given the current lack of understanding regarding human SSCs regulatory mechanisms. Notably, in this study, we observed clear co-localization of MTMR7 with the established SSCs marker PLZF in adult human testis tissue via immunofluorescence staining, providing additional evidence that MTMR7 is indeed expressed in human SSCs at the protein level.

Furthermore, considering the known species differences between mouse and human SSCs in terms of cell types, markers, and phenotypes, we felt it was important to directly explore the potential function of MTMR7 in human SSCs. Our aim was to contribute novel insights into human-specific SSCs biology and address gaps in the current literature. We sincerely appreciate your comment, which has prompted us to further clarify and highlight these considerations in the revised Introduction section.

1.Introduction:

The introduction provides a good background on the importance of SSCs in spermatogenesis and male fertility. However, it could benefit from a more detailed discussion on the differences between mouse and human SSCs, particularly regarding the molecular pathways involved. This would help contextualize the significance of studying MTMR7 in human SSCs.

Suggestion: Expand the discussion on the differences in molecular mechanisms between mouse and human SSCs to better justify the need for this study.

Response: Thank you for this valuable suggestion. We fully agree that clarifying the differences between mouse and human SSCs would strengthen the rationale for focusing on human SSCs in this study. Accordingly, we have expanded the Introduction section to include a more detailed discussion of these interspecies differences. The revised paragraph is as follows:

“However, the molecular mechanisms controlling the development of human SSCs remain largely unknown and differ significantly from those in mice [16]. For instance, in addition to the commonly expressed markers PLZF and GPR125, human SSCs uniquely express MAGE-A4 and CD133, while mouse SSCs are marked by CD29, KIT, and POU5F1 [17, 18]. The CXCR4–CXCL12 signaling pathway, essential for SSCs homing to the niche, also differs between species: in humans, CXCL12 is mainly expressed by Leydig cells, whereas in mice, it is predominantly produced by Sertoli cells [19]. These differences underscore the need for species-specific studies to elucidate the regulatory mechanisms governing human SSCs function, which is essential for developing targeted therapies for male infertility.”

2.Materials and Methods:

The use of si-RNAs and overexpression plasmids is well described, but the authors should provide more information on the efficiency and specificity of these tools. For example, were off-target effects of si-RNAs assessed?

Suggestion: Clearly specify the sample size for each experiment and provide additional details on the efficiency and specificity of the si-RNAs and overexpression plasmids used.

Response: Thanks again for this suggestion. We have revised the Materials and Methods section to provide additional details regarding the design, validation, and efficiency of the siRNAs and overexpression plasmids used in our study. We also clarified the measures taken to minimize potential off-target effects. The revised paragraph is as follows:

“When the cell density reached approximately 70%, overexpression plasmids pcDNA3.1-Flag-MTMR7 and an empty vector (Sangon, Shanghai, China) were introduced into SSCs using the X-treme GENE HP DNA transfection reagent (Mannheim, Germany). For plasmid transfections, 10 μg DNA was used per 10-cm dish and 2 μg per well of a 6-well plate. Separately, siRNAs targeting MTMR7 and FLNB (GenePharma, Suzhou, China) were transfected into SSCs using Lipofectamine 2000 (Invitrogen, USA), with 10 μl siRNAs per well in a 6-well plate [40-42]. To minimize off-target effects, two independent siRNAs were designed for each gene, and only consistent results from both were considered valid. The sequences targeted by the siRNAs were as follows:

si-MTMR7-#1: 5′-GCGAUGUGAAUAGAGACUA-3′.

si-MTMR7-#2: 5′-CGGCCUAAACUUAAUGCAA-3′.

si-FLNB-#1: 5′-GCACGGUCACUGUUAGAUA-3′.

si-FLNB-#2: 5′-GAUCGUGUGAUGUCAAAUA-3′.

si-NC: 5′-UUCUCCGAACGUGUCACGU-3′.

The knockdown efficiency of all siRNAs was validated by qPCR prior to use. Transfection efficiency for both overexpression plasmids and siRNAs was further confirmed by Western blot analysis. The primers used for qPCR ar

---

## [Decision Letter · Decision Letter 1]

MTMR7 regulates human spermatogonial stem cells proliferation and migration via targeting FLNB

PONE-D-24-58024R1

Dear Dr. Shen,

We’re pleased to inform you that your manuscript has been judged scientifically suitable for publication and will be formally accepted for publication once it meets all outstanding technical requirements.

Kind regards,

Jeevithan Elango, PhD

Academic Editor

PLOS ONE

Additional Editor Comments (optional):

The manuscript is acceptable in the present form.

Reviewers' comments:

Reviewer's Responses to Questions

**Comments to the Author**

1. If the authors have adequately addressed your comments raised in a previous round of review and you feel that this manuscript is now acceptable for publication, you may indicate that here to bypass the “Comments to the Author” section, enter your conflict of interest statement in the “Confidential to Editor” section, and submit your "Accept" recommendation.

Reviewer #2: All comments have been addressed

2. Is the manuscript technically sound, and do the data support the conclusions?

Reviewer #2: Yes

3. Has the statistical analysis been performed appropriately and rigorously? 

Reviewer #2: Yes

4. Have the authors made all data underlying the findings in their manuscript fully available?

Reviewer #2: Yes

5. Is the manuscript presented in an intelligible fashion and written in standard English?

Reviewer #2: Yes

6. Review Comments to the Author

Reviewer #2: (No Response)

7. PLOS authors have the option to publish the peer review history of their article (what does this mean? ). If published, this will include your full peer review and any attached files.

**Do you want your identity to be public for this peer review?** For information about this choice, including consent withdrawal, please see our Privacy Policy .

Reviewer #2: **Yes: ** Dai Zhou

---

## [Editor Report · Acceptance letter]

PONE-D-24-58024R1

PLOS ONE

Dear Dr. Shen,

I'm pleased to inform you that your manuscript has been deemed suitable for publication in PLOS ONE. Congratulations! Your manuscript is now being handed over to our production team.

Kind regards,

on behalf of

Dr. Jeevithan Elango

Academic Editor

PLOS ONE